# Systematic Review and Meta-Analysis of the Usefulness of Epicardial Fat Thickness as a Non-Invasive Marker of the Presence and Severity of Nonalcoholic Fatty Liver Disease

**DOI:** 10.3390/biomedicines10092204

**Published:** 2022-09-06

**Authors:** Lorenzo A. Orci, François R. Jornayvaz, Christian Toso, Karim Gariani

**Affiliations:** 1Division of Abdominal and Transplantation Surgery, Geneva University Hospitals, University of Geneva, 1205 Geneva, Switzerland; 2Division of Digestive Surgery, Department of Surgery, Centre Hospitalier de l’Université de Montreal (CHUM), Montreal, QC H2X 0C1, Canada; 3Division of Endocrinology, Diabetes, Nutrition and Therapeutic Patient Education, Department of Medical Specialties, Geneva University Hospitals, 1211 Geneva, Switzerland; 4Diabetes Center, Faculty of Medicine, University of Geneva, 1211 Geneva, Switzerland; 5Department of Cell Physiology and Metabolism, Centre Medical Universitaire (CMU), Faculty of Medicine, University of Geneva, 1211 Geneva, Switzerland; 6Hepato-Pancreato-Biliary Center, Geneva University Hospitals, 1211 Geneva, Switzerland

**Keywords:** non-alcoholic fatty liver disease, steatosis, epicardial fat, non-invasive marker, CT-SCAN, MRI, ultrasonography

## Abstract

We performed a systematic review and meta-analysis to assess the association between epicardial fat thickness (EFT) and nonalcoholic fatty liver disease (NAFLD). This systematic review was conducted in accordance with the Preferred Reporting Items for Systematic reviews and Meta-Analysis (PRISMA) and was based on a registered protocol (CRD 4201809 5493). We searched Medline and Embase until December 2021 for studies reporting on the association between EFT and NAFLD. Qualitative reviews, meta-analyses and meta-regressions were performed to explore this association. Effect sizes are reported as standardized mean differences. We included 12 studies, comprising 3610 individuals. EFT was evaluated with trans-thoracic echocardiography in nine studies, two studies using cardiac computed tomography and one study using magnetic resonance imaging (MRI). The presence of NAFLD was evaluated using transabdominal liver ultrasound in nine studies. Other studies used histology, magnetic resonance spectroscopy and MRI-derived proton density fat fraction. Liver biopsy was performed to assess the severity of NAFLD in four studies. The random-effects meta-analysis indicated that, as compared to control patients with lean livers, patients with NAFLD displayed significantly higher EFT (standardized mean difference 0.61, 95% confidence interval: 0.47–0.75, *p* < 0.0001, I^2^ = 72%). EFT was further significantly higher in patients with severe liver steatosis versus patients with mild–moderate liver steatosis (standardized mean difference 1.21 95% confidence interval: 0.26–2.16, *p* < 0.001, I^2^ S = 96%). Through the meta-regression analysis, we found that patients with increasingly higher blood levels of aspartate aminotransferase displayed an increasingly higher depth of association. The current meta-analysis suggests that EFT may represent a useful surrogate for assessing the presence and severity of NAFLD in a non-invasive manner.

## 1. Introduction

Epicardial adipose tissue is a visceral fat deposit positioned between the heart and the pericardium. Epicardial fat covers around 80% of the surface of the heart, representing 20% of the organ’s total weight [1]. Putative roles of epicardial adipose tissue comprise cardiac thermoregulation, lipid storage, control of coronary artery vasomotion and coronary atherosclerosis development [2,3]. In clinical practice, excessive epicardial fat is associated with left ventricular dysfunction, coronary artery disease and atrial fibrillation [4,5,6]. Epicardial fat thickness can be measured using non-invasive radiological approaches such as transthoracic echocardiography, computerized tomography or magnetic resonance imaging (MRI, gold standard) [7]. The growing use of thoracic imaging has contributed to an increase in the detection of epicardial fat deposits, raising interest as to whether this adipose tissue may be a marker for cardiovascular disease or for metabolic syndrome.

Nonalcoholic fatty liver disease (NAFLD) is defined as excessive lipid accumulation in the liver in the absence of secondary causes of hepatic fat accumulation such as alcohol consumption (<20 g/day in men and 10 g/day in women), viral hepatitis, use of steatogenic medication, or hereditary disorders [8]. NAFLD is now the most common cause of chronic liver disease worldwide and it is estimated that up to 25% of the global population may be affected by this condition [8,9]. Abundant evidence has documented the role of NAFLD as an independent cardiovascular disease risk factor, and indeed cardiovascular events are the main cause of mortality among patients with NAFLD [10,11,12].

Liver biopsy is the gold standard for the diagnostic evaluation of the severity of NAFLD [8], but this procedure remains invasive and can lead to complications including pain, as well as minor or even major bleeding [13].

To avoid such pitfalls, non-invasive approaches to assess liver fibrosis have been developed, using a combination of clinical–biochemical factors and imaging techniques. Simple non-invasive tools to predict liver fibrosis and liver-related morbidity include the NAFLD Fibrosis Score (NFS) and Fibrosis-4 (FIB-4) score that are endorsed by the European Association for the Study of the Liver (EASL), European Association for the Study of Diabetes (EASD) and European Association for the Study of Obesity (EASO) clinical practice guidelines [14]. These scores appear to be reliable, easy to perform and have shown good ability to stratify the risk of liver-related morbidity and mortality. Some authors even consider such risk-scoring tools to offer similar performances to a liver biopsy [15]. When using the FIB-4 score, patients falling in the low-risk categories of liver-related morbidity usually require no additional investigation, but clinical and biological follow-up should be considered. When risk estimations are intermediate or high, a transient elastography should be performed. Transient elastography represents the most validated imaging tool to assess the stage of fibrosis in NAFLD patients and it allow a stratification for the potential use of a liver biopsy [8]. In this regard, a meta-analysis based on 19 studies and 2495 NAFLD patients showed that transient elastography had remarkable accuracy for diagnosing advanced fibrosis with a summary AUROCS of 0.88 [16]. Despite these encouraging lines of evidence, the routine use of non-invasive NAFLD markers remains limited, especially outside the specialized practice of hepato-gastroenterology, and NALFD remains a largely underdiagnosed condition. In this regard, novel surrogate markers for NAFLD and its severity are or interest [17]. 

Both epicardial fat thickness and liver steatosis can be considered as features of the metabolic syndrome in that they both correspond to the accumulation of ectopic lipids. One could therefore hypothesize their presence to be concomitant. Several studies have addressed this question [18,19,20]. However, the evidence remains inconclusive, in part due to the small sample sizes of the individual studies and due to the presence of potential confounders. We therefore embarked on this systematic review of the literature to assess the association between epicardial fat thickness and the presence and severity of NAFLD.

## 2. Method

### 2.1. Data Sources and Searches

The search strategy, study selection, data extraction and analysis were performed according to a predefined protocol described in the International Prospective Register of Systematic Reviews (PROSPERO, CRD 42014013578). This systematic review was conducted according to Preferred Reporting Items for Systematic reviews and Meta-Analysis (PRISMA) guideline [21].

We gathered all studies providing a quantitative estimate of the association between EFT and NAFLD presence and severity. The literature search was completed using Medline (1966–December 2021) and EMBASE (1980–December 2021). Within these databases, we conducted structured computerized literature searches using the following search terms: «((nonalcoholic steatohepatitis) OR (non-alcoholic fatty liver disease)) AND ((epicardial fat) OR (epicardial adipose tissue))». Duplicate entries were excluded. No time limit or language restrictions were applied to the search. Reference lists of retrieved articles were hand-searched to identify additional relevant articles. Two investigators (KG and LO) reviewed the titles, abstracts, and full-text articles. Discrepancies concerning study inclusion were resolved by discussion between the two authors or with a third author (FRJ). 

### 2.2. Data Extraction and Methodological Quality Rating

We retrieved data on patient age, gender, and on the imaging techniques used to characterize epicardial fat thickness and NAFLD. We used the Newcastle–Ottawa Scale (NOS) to rate the methodological quality of the included studies [22]. The NOS is composed of three sections: patient selection (up to four points), between-groups comparability (up to two points), and outcome definition and reporting (up to three points). The maximum score given by summing the value of each section is nine points. Based on this calculation, the methodological quality of each study was considered as poor (score, 0–3), fair (score, 4–6), or good (score, 7–9). Studies with overly severe flaws (NOS ≤ 2) were excluded.

### 2.3. Data Synthesis and Analysis

Because of the various units of measures reported in the individual studies, the primary endpoint of the current meta-analysis was the standardized mean difference in epicardial fat between patients with or without NAFLD. We further compared epicardial fat thickness among (i) patients with severe steatosis versus those with mild–moderate steatosis and (ii) patients with advanced (F3–F4) liver fibrosis versus those with early (F1–F2) fibrosis (when results of a liver biopsy were reported). Random-effects models were used. We did meta-regression analyses to assess the relationship between the calculated effect sizes and patient blood level of aminotransferases. Statistical heterogeneity of the results was assessed with the Cochrane Q test and the I^2^ value. Analyses were performed with Revman 5.3 (Cochrane Collaboration) and Stata 15 (Stata Corp, College Station, TX, USA).

## 3. Results

### 3.1. Characteristics of Included Studies, Methodological Quality

Our search strategy identified 146 unique publications, of which 121 were excluded on the basis of their title and abstract. We examined 25 full-text citations. A total of 12 studies were eventually retained in our systematic review. Overall, data on 3610 patients were gathered. The study selection process is illustrated in Figure 1. Nine studies took place in Europe, two in Asia and one in North America. There were seven case–control studies, four cross-sectional studies and one prospective cohort study.

The presence of NAFLD was evaluated with ultrasound in nine studies. Magnetic resonance (MR)-imaging-derived proton density fat fraction, MR spectroscopy and liver biopsy were used in one study each, respectively. To quantify epicardial fat thickness, nine studies used trans-thoracic echocardiogram (TTE), one study used MR imaging and two studies used cardiac computed tomography (CT) (Table 1). As for the relationship between epicardial fat thickness and the severity of liver steatosis, six studies provided comparisons of epicardial fat thickness in patients with severe versus mild or moderate liver steatosis [6,18,20,23,24,25] and two studies compared individuals with severe (F3–F4) versus early (F1–F2) hepatic fibrosis [18,23]. 

Critical appraisal of the included studies is shown in Table 1. Based on the NOS scale, one study scored a nine, ten studies scored a seven, and one study scored a six. No study was excluded due to low quality (score of less than two).

### 3.2. Quantitative Assessment

The primary outcome of interest of the current meta-analysis was the standardized mean difference in epicardial fat thickness between patients with or without NAFLD (Figure 2). A random-effects meta-analysis of the 12 included studies showed a strongly significant difference between groups (standardized mean difference 0.61, moderate effect size, 95% CI: 0.47–0.75, *p* < 0.001), suggesting that epicardial fat thickness was greater in patients with NAFLD. We observed a significant heterogeneity across studies (I^2^ = 72%, Q test *p* < 0.1). 

Six studies compared epicardial fat thickness between patients with severe liver steatosis versus those with only mild–moderate steatosis. Though being highly heterogeneous (I^2^ = 96%), the results indicated that patients with severe liver steatosis display a markedly higher epicardial fat thickness as compared to their leaner counterparts (standardized mean difference 1.21 95% CI: 0.26–2.16, *p* = 0.010) (Figure 3). Furthermore, two studies compared epicardial fat thickness according to the presence of severe (F3–F4) versus early (F1–F2) liver fibrosis. Consistent with the above results, a meta-analysis revealed that patients with advanced liver fibrosis harbored significantly higher amounts of epicardial fat as compared to patients with early fibrosis (standardized mean difference 0.66 95% CI: 0.35–0.97, *p* < 0.001, no significant heterogeneity) (Figure 4). 

By using a meta-regression analysis, we found a significant correlation between the calculated effect size and patient blood level of aspartate aminotransferase (*p* = 0.024). The trend was similar for alanine aminotransferase, though this meta-regression did not reach statistical significance (Figure 5A,B).

## 4. Discussion

In this meta-analysis of non-randomized studies the pooling data of 3610 patients, we found evidence supporting that the presence of exaggerated epicardial adipose tissue is associated with NAFLD. The vast majority of the studies included in this systematic review consistently showed that epicardial fat was more abundant in patients with NAFLD as compared to the control patients. This association is further supported by the results of two sensitivity analyses, which showed incrementally greater amounts of adipose tissue in patients with severe fatty liver infiltration, and a significant difference between patients with advanced vs. early liver fibrosis. Meta-regression analysis also revealed a significant and positive correlation between the depth of association reported by the individual studies and baseline AST levels.

A single study did not find a positive association between epicardial fat and the presence of NAFLD [26]. As mentioned by the authors, the control group included individuals with increased levels of aminotransferases and γ-GT but who did not show fatty liver infiltration during liver ultrasound. One can therefore not rule out that, in this study, the echographic assessment was not sensitive enough to detect some cases of NAFLD in this group, leading to the misclassification of cases. It is noteworthy that the authors observed a positive association between epicardial fat thickness and markers of inflammation and insulin resistance, both being common findings in NAFLD [9]. 

The interpretation of a standardized mean difference as an effect size may be challenging in clinical practice [27] and the clinical relevance of this measure has been a matter of debate for decades [28]. For instance, Cohen [29] proposed interpreting standardized mean difference values of 0.2, 0.5, and 0.8 as corresponding to, respectively, a small, moderate, and large effect size. Using this approach, our results would convey the message that epicardial fat thickness may have a greater role at discriminating between patients with marked vs. limited liver steatosis (standardized mean difference 1.21, 95% CI: 0.26–2.16) than merely at trying to detect NAFLD (standardized mean difference 0.61, 95% CI: 0.47–0.75). Another approach to interpreting standardized mean differences would be to use the cut-off value of 0.5 to consider a result of clinical significance. In this case, as well, results of the current meta-analysis indicate that epicardial fat measurement may be of value in the clinical context [30]. 

A fine-tuned coordination between various organs and tissues is required for the maintenance of systemic homeostasis [31]. The accumulation of fat in multiple organs is observed in several conditions, such as metabolic syndrome, alcoholic liver disease, viral hepatitis, or chemotherapy-induced parenchymal lesions, and contributes to the development or aggravation of pathologies such as atherosclerosis or diabetes [32,33,34,35]. However, the mechanisms underlying these observations remain only partially elucidated and are currently an active field of investigation. Ectopic fat deposits such as visceral fat or dorso-cervical fat have already been shown to play a role in the pathogenesis of NAFLD and in its histological severity [36,37]. The association between NAFLD and epicardial fat thickness may explain at least in part the increased risk of cardiovascular risk among patients with NAFLD [12]. Elevated amounts of epicardial adipose tissue may not only be a maker of a hepatic involvement in the metabolic syndrome, but also of other organs such as blood vessels or the heart itself. In this regard, it has been shown that epicardial fat thickness can independently predict coronary artery atherosclerosis plaque vulnerability [34,38]. Similarly, several lines of evidence suggest that epicardial adipose tissue deposits and the associated cardiovascular risk may be reduced through physical exercise, weight loss and pharmacological interventions with glucagon-like peptide 1 receptor agonists and sodium-glucose co-transporter 2 inhibitors [39,40,41,42]. Further longitudinal studies are needed to assess whether epicardial fat thickness may be an independent predictor of the risk of progressive cardiovascular disease among NAFLD patients, especially considering that cardiovascular events remain the first cause of mortality in this population.

It is noteworthy that our work has several limitations. First, our systematic review and meta-analysis included observational studies that by definition carry systematic biases or undetected confounding factors. Such methodological issues limit us from making causal inferences on whether epicardial adipose tissue and NAFLD are both concomitant phenotypes that arise in patients with metabolic syndrome, or whether one of them precedes the other. Another limitation is that the majority of included studies used liver ultrasound to diagnose NAFLD instead of liver histology, which may have resulted in some degree of misclassification. However, liver ultrasound is the first-line diagnostic modality for NAFLD as proposed by the American College of Gastroenterology and the American Gastroenterology Association [43]. On a similar note, most of the included studies used transthoracic echocardiography to assess epicardial fat thickness, while the gold-standard technique in this context is MRI [44]. One should bear in mind that even though transthoracic echocardiography is less costly and more readily available, it remains less accurate than MRI or CT and carries the drawbacks of being operator-dependent [7] and requiring clear acoustic windows, which can prove to be challenging in obese patients [45]. In addition, transthoracic echocardiography uses linear measurements of epicardial fat thickness (while mediastinal adiposity forms three-dimensional conglomerates). Another limitation to our work is that the included studies reported only crude associations between epicardial fat thickness and liver steatosis but did not report dedicated measures of diagnostic accuracy such as sensitivity, specificity, and positive predictive values.

In conclusion, while awaiting more prospective data, the currently available evidence indicates that excessive amounts of epicardial fat may be associated with NAFLD, the degree of liver steatosis, and the severity of liver fibrosis. Future research should identify reproducible thresholds of epicardial fat thickness and evaluate the cost-effectiveness of this approach. Comparing the impact of epicardial fat to that of other forms of ectopic adipose tissue would also address an important gap in the knowledge.

## Figures and Tables

**Figure 1 biomedicines-10-02204-f001:**
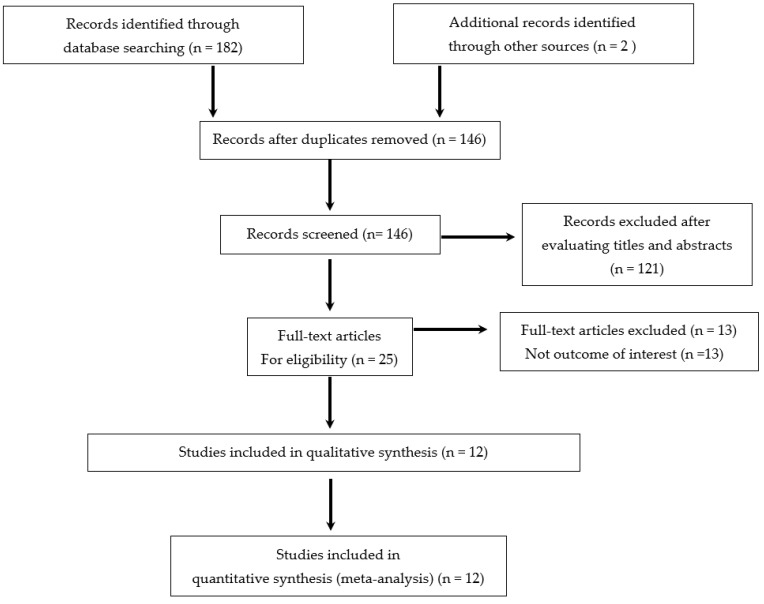
Flow chart of the inclusion/exclusion process, according to the Preferred Reporting Items for Systematic reviews and Meta-Analysis (PRISMA) guidelines.

**Figure 2 biomedicines-10-02204-f002:**
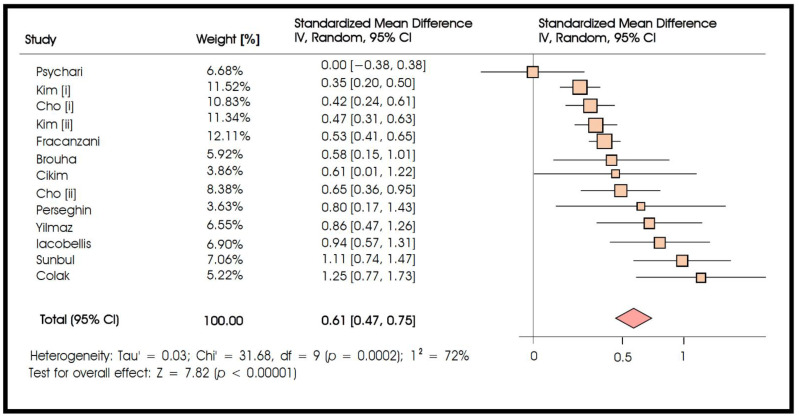
Meta-analysis of studies comparing epicardial fat thickness in NAFLD vs. control patients.

**Figure 3 biomedicines-10-02204-f003:**
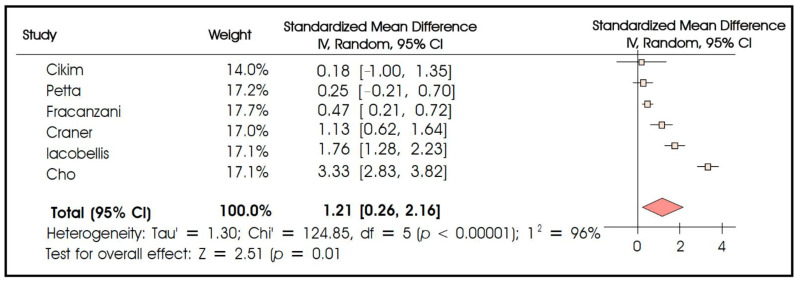
Meta-analysis of studies comparing epicardial fat thickness in patients with severe versus mild-to-moderate liver steatosis.

**Figure 4 biomedicines-10-02204-f004:**
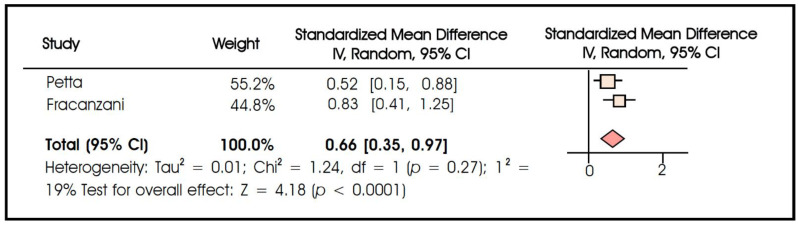
Meta-analysis of studies comparing epicardial fat thickness in patients with severe (F3–F4) versus early (F1–F2) liver fibrosis.

**Figure 5 biomedicines-10-02204-f005:**
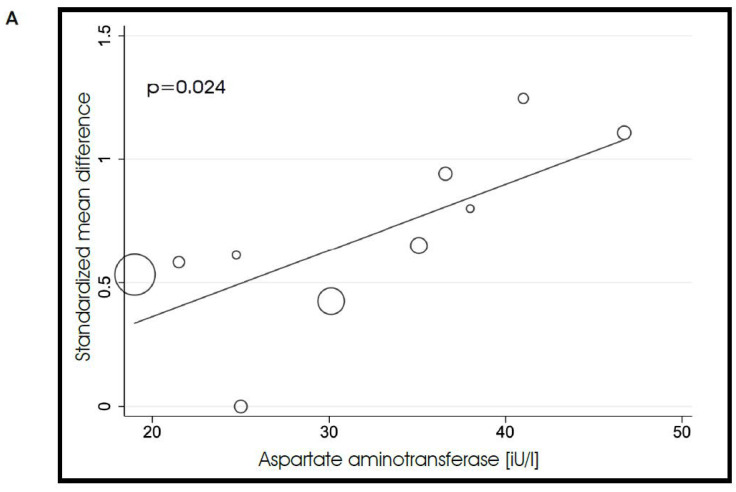
Meta-analysis regression looking at the impact of baseline aspartate (**A**) and alanine (**B**) aminotransferase levels on the calculated effect size.

**Table 1 biomedicines-10-02204-t001:** Main characteristics of included studies.

Author, Year	Design	Country	NAFLD (n)	Control (n)	Age (yr) NAFLD	Age (yr) Control	Male NAFLD (%)	Male Control (%)	Method for NAFLD Diagnosis and/or Intrahepatic Lipid Measurement	Liver Biopsy Performed for NAFLD Severity Assessment	Method for Epicardial Fat Thickness Measurement	NOS Score
Brouha, 2017	Cross-sectional	USA	62	33	57.4 ± 11.1	62.6 ± 10.3	51.6	57.6	MRI-PDFF	N	CT	6
Cho, 2017	Case-control	South Korea	309	463	51.9	52.6	88.0%	71.1	Ultrasound	N	TTE	7
Cikim, 2007	Cross-sectional	Turkey	49	14	-	-	0	0	Ultrasound	N	TTE	7
Colak, 2012	Case-control	Turkey	57	30	44.2 ± 9.4	42.7 ± 14.5	83.9	87.5	Ultrasound and liver biopsy	Y	TTE	7
Fracanzani, 2016	Cross-sectional	Italy	512	0	61 ± 13	-	61.3	-	Ultrasound and liver biopsy	Y	TTE	7
Iacobellis, 2014	Prospective cohort study	Italy	62	62	43.9 ± 9.3	44 ± 8.5	67.8	72.6	Ultrasound	N	TTE	9
Kim, 2016	Case-control	South Korea	676	796	44.0 ± 8.0	44.3 ± 9.1	94.0	76.1	Ultrasound	N	CT	7
Oguz, 2016	Case-control	Turkey	41	37	37.9 ± 8.9	34.5 ± 8.6	65.9	46.0	Ultrasound	N	TTE	7
Perseghi, 2007	Case-control	Italy	21	21	35 ± 7	36 ± 7	100	100	^1^H-MRS	N	MRI	7
Psychari, 2016	Cross-sectional	Greece	57	48	50 ± 13	50 ± 15	61	56	Ultrasound	N	TTE	7
Sunbul, 2014	Case-control	Turkey	100	50	44.8 ± 9.8	45.1 ± 6.3	59	68	Ultrasound and liver biopsy	Y	TTE	7
Yilmaz, 2011	Case-control	Turkey	54	56	47 ± 10	46 ± 11	48.1	48.2	Liver biopsy	Y	TTE	7

CT: Computed tomography; ETT: Transthoracic echocardiogram; ^1^H-MRS: Proton magnetic resonance spectroscopy; MRI-PDFF: Magnetic Resonance Imaging Proton Density Fat Fraction.

## Data Availability

Not applicable.

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
