# Peer review of "Systematic Review and Meta-Analysis of the Usefulness of Epicardial Fat Thickness as a Non-Invasive Marker of the Presence and Severity of Nonalcoholic Fatty Liver Disease"

_biomedicines, 2022, doi:10.3390/biomedicines10092204_

Round 1

Reviewer 1 Report

To Karim Gariani,

The manuscript is of interest to the scientific and medical communities, as it highlights the relevance of non-invasive methods for NAFLD diagnosis. The design of the synthesis is adequate and results are consistent and accompanied by a good discussion. Thus I have recommended to accept it after minor revision.

The main concern is the quality of the figures and the letter, as sometimes a different font is used in each part of the text.

Author Response

We thank the reviewer for his/her comments. We have modified the figures to improve the quality, in particular figure 1 the flow chart. We have used now the same font in each part of the text.

Reviewer 2 Report

This is a systematic review report written by Orci et al. concerning use of epicardial fat thickness (EFT) as a surrogate marker of NAFLD. Their firm methodology as meta-analysis led sufficiently reliable results. However, I doubt the clinical significance of this report. If the study had aimed on replacement of an invasive test, liver biopsy, to noninvasive test, EFT, the results might have suggested clinical implication. Moreover, a similar study was already published (Liu B, et al. Hepatol Int 2019). I regret to give such a negative comment.

Author Response

We thank the reviewer for his/her comments. We agree that clinical implementation of EFT to detect NAFLD remain to be assessed and yet it could not replace invasive test such as liver biopsy, however our data suggest that it has to be considered in the future. Regarding a study with similar approach already published, our work included in part other studies and in addition we performed additional analysis such meta-regression to look at the impact of baseline aspartate and alanine aminotransferase levels. As suggested, we used an English editing service to improve the English language.

Reviewer 3 Report

It is an excellent article on an interesting topic. My only suggestion might be to ask for a native english speaker to revise english language and grammar to improve readability.

Author Response

We thank the reviewer for his/her positive comments. We have used the service of an English editing service to correct the grammar in order to improve the readability of the manuscript.

Round 2

Reviewer 2 Report

Unfortunately, substantial improvement can't be seen in the revised version of the manuscript.